# Normalizing Flows for Out-of-Distribution Detection: Application to Coronary Artery Segmentation

Costin Florian Ciușdel [1,2,*], Lucian Mihai Itu [1,2], Serkan Cimen [3], Michael Wels [4], Chris Schwemmer [4], Philipp Fortner [5], Sebastian Seitz [5], Florian Andre [5], Sebastian Johannes Buß [5], Puneet Sharma [3] and Saikiran Rapaka [3]

1   Advanta, Siemens SRL, 15 Noiembrie Bvd, 500097 Brasov, Romania; lucian.itu@siemens.com
2   Automation and Information Technology, Transilvania University of Brasov, Mihai Viteazu nr. 5, 500174 Brasov, Romania
3   Digital Services, Digital Technology & Innovation, Siemens Healthineers, 755 College Road, Princeton, NJ 08540, USA; serkan.cimen@siemens-healthineers.com (S.C.); sharma.puneet@siemens-healthineers.com (P.S.); saikiran.rapaka@siemens-healthineers.com (S.R.)
4   Computed Tomography-Research & Development, Siemens Healthcare GmbH, 91301 Forchheim, Germany; michael.wels@siemens-healthineers.com (M.W.); chris.schwemmer@siemens-healthineers.com (C.S.)
5   Das Radiologische Zentrum-Radiology Center, 74889 Sinsheim, Germany; dr.fortner@das-radiologische-zentrum.de (P.F.); dr.seitz@das-radiologische-zentrum.de (S.S.); florian.andre@med.uni-heidelberg.de (F.A.); prof.buss@das-radiologische-zentrum.de (S.J.B.)
*   Correspondence: costin.ciusdel@unitbv.ro

**Abstract:** Coronary computed tomography angiography (CCTA) is an effective imaging modality, increasingly accepted as a first-line test to diagnose coronary artery disease (CAD). The accurate segmentation of the coronary artery lumen on CCTA is important for the anatomical, morphological, and non-invasive functional assessment of stenoses. Hence, semi-automated approaches are currently still being employed. The processing time for a semi-automated lumen segmentation can be reduced by pre-selecting vessel locations likely to require manual inspection and by submitting only those for review to the radiologist. Detection of faulty lumen segmentation masks can be formulated as an Out-of-Distribution (OoD) detection problem. Two Normalizing Flows architectures are investigated and benchmarked herein: a Glow-like baseline, and a proposed one employing a novel coupling layer. Synthetic mask perturbations are used for evaluating and fine-tuning the learnt probability densities. Expert annotations on a separate test-set are employed to measure detection performance relative to inter-user variability. Regular coupling-layers tend to focus more on local pixel correlations and to disregard semantic content. Experiments and analyses show that, in contrast, the proposed architecture is capable of capturing semantic content and is therefore better suited for OoD detection of faulty lumen segmentations. When compared against expert consensus, the proposed model achieves an accuracy of 78.6% and a sensitivity of 76%, close to the inter-user mean of 80.9% and 79%, respectively, while the baseline model achieves an accuracy of 64.3% and a sensitivity of 48%.

**Keywords:** out-of-distribution; normalizing flows; coronary computed tomography angiography; lumen segmentation

## 1. Introduction

Coronary computed tomography angiography (CCTA) is an effective imaging modality, increasingly accepted as a first-line test to diagnose coronary artery disease (CAD). Advancements in CCTA have allowed for minimal radiation exposure, effective coronary characterization, and detailed imaging of atherosclerosis over time. Due to the increasing body of evidence showing the effectiveness of CCTA [1,2], recent ACC/AHA chest pain guidelines recommend CCTA as a first line test for patients with stable and acute chest pain.

The rapid progress in Artificial Intelligence (AI) approaches for pattern recognition over the last decade has led to several concepts, applications, and products built around the

primary goal of augmenting and/or assisting the radiologist in their reading and reporting workflow, mainly focusing on automatic detection and characterization of features and on automatic measurements in the images. The majority of state-of-the-art CCTA image analysis algorithms are powered by artificial intelligence (AI) [3]. The accurate segmentation of the coronary artery lumen on CCTA is a crucial step for the automated detection and assessment of CAD for numerous use cases:

- Anatomical quantification of stenosis—for instance, minimum lumen area, minimum lumen diameter or percentage diameter stenosis [4].
- Morphological quantification: amount and composition of coronary plaques [5].
- Functional quantification of coronary function—for instance, CFD or machine learning based Fractional Flow Reserve (FFR) computation [6,7].

While the performance of AI based methods has improved markedly over the years, given the importance of an accurate lumen detection, semi-automated approaches are currently still being employed. Thus, the lumen is first automatically detected, and then manually inspected and edited by the radiologist if deemed necessary. This process, together with coronary artery centerline editing, required, e.g., between 10 and 60 min in a study assessing the diagnostic performance of ML-based CT-FFR for the detection of functionally obstructive coronary artery disease [6]. One potential approach for significantly reducing the time required for a semi-automated CCTA lumen analysis is to pre-select locations which are likely to require inspection and editing, and to present only those for review to the radiologist. Considering that a Deep Neural Network (DNN) is responsible for generating the lumen segmentation masks, this pre-selection step can be linked to the topic of confidence and out-of-distribution detection in Deep Learning. It is known that the output of classic DNNs may be unreliable when applied on out-of-domain, noisy or uncertain input data. Many methods have been proposed for assessing model output confidence. van Amersfoort et al. [8] uses a bi-Lipschitz deep feature extractor which feeds a sparse Gaussian Process (GP): segmenting an image involves at the lowest level many classification sub-problems, where each pixel is labelled according to the object/class it pertains to. Therefore, a GP can be adapted to model a segmentation task as a classification task, and, hence, the associated output uncertainty value can be extracted for each pixel. Image segmentations which display large mask uncertainties can be flagged as unreliable and proposed for human inspection. Another approach is to employ energy-based models. Liu et al. [9] shows that a model trained with a SoftMax final activation contains implicitly a density estimator. An energy-score can be computed for each pixel and aggregated mask-scores can be compared to predefined thresholds to determine which samples require manual inspection. Within these two methods, the model confidence is shaped during learning the target task. Therefore, in classification problems, the output confidence is low whenever the input sample is either far away from the training distribution or it is placed close to the nonlinear class-separation manifold in the input space.

Regular confidence methods do not provide a reason why the output confidence is low, and the class separation learnt by the model is highly dependent on the target task and on the model architecture. Normalizing Flow (NF) models on the other hand can be trained explicitly to model input data probability densities. Given a downstream target task $T$, if only its input data is employed for building the NF model, then estimating the likelihood of input samples for the target task can be obtained through the NF model. Input samples with low probabilities can be flagged as out-of-distribution and the target model's output should be considered unreliable, as it would operate outside its training distribution. An NF model can also be built by stacking the input samples with their expected GT output. This way, the NF model can be placed downstream of the target task and act as an Audit model, detecting cases where the previous model provided faulty predictions. In either scenario, the NF is a separate model and therefore imposes no constraints on the model responsible for the target task. NFs are a class of generative models which can perform exact log-likelihood computation. They have been employed in various setups, for instance:

- Image generation[10,11]: Being reversible models, random samples from the prior distribution can be transformed into the data domain, therefore obtaining new synthetic datapoints.
- Prior for variational inference: Instead of employing a fixed distribution (usually the normal distribution) in the KL term for ELBO maximization in variational inference, NF can be employed to model a much more expressive prior distribution. In a variational auto-encoder (VAE), this allows the encoder to better capture input patterns by not placing a fixed constraint on its computed embeddings. Ziegler and Rush [12] employed such a method for character-level language modeling and polyphonic music generation.
- Out-of-Distribution (OoD) detection: As log-likelihood values can be exactly and efficiently computed, NF may be good candidates in outlier detection [13].

NF can usually operate efficiently in both directions: forward (or inference) direction, where input samples $x$ from the input domain $X$ are transformed into embeddings $z$ which are likely under a chosen distribution $Z$. At each layer, the input is modified towards $Z$ and the *logDet* value (i.e., $\ln\left(\left|\det\left(\frac{\partial f}{\partial x}\right)\right|\right)$, where $f$ is the NF) is summed with the current layer contribution. The backward (or generative) direction employs the bijection property of the NF to transform an embedding $z$ into a synthetic sample $x_{new}$. Refs. [14,15] offer an introduction and review into the current approaches used in the NF framework.

In this paper, we present an approach based on NF for the OoD detection of coronary lumen segmentations. NF models which are built from coupling layers as proposed in [10,11] tend to focus on local pixel correlations instead of the global semantic meaning [13,16] and, as a result, OoD samples may in fact produce larger log-probability values than in-distribution data. We investigate the usage of a new type of coupling layer, which employs reversible $1 \times 1$ convolutions in which the filter parameters are computed based on the passed-through components. We compare the proposed architecture against a Glow-like architecture on the task of detecting mismatched pairs of CCTA lumen images and their corresponding lumen segmentations. The coronary lumen images and masks are 3D volumes stacked along the channel axis. We also employ synthetic perturbations on the binary masks and use the perturbed samples as explicit outliers to further shape the learnt probability density of "correct" image-mask pairs. The end goal is to flag those samples for which the given segmentation does not properly match with the lumen image. Overall, we assess the performance of the NF models as follows: (i) against the synthetic mask perturbations, and (ii) using expert annotations.

## 2. Methods

### 2.1. Patients and Imaging Protocol

Two datasets were used for the purpose of this study: a primary dataset as basis for the conventional train\validation\test split and an additional secondary separate test set.

The primary dataset included 560 patients who underwent contrast enhanced CCTA for clinical indications at Das Radiologische Zentrum (Heidelberg, Germany). CCTA was performed on a third generation dual-source CT scanner (SOMATOM Force, Siemens Healthcare GmbH, Erlangen, Germany). Beta-blockers or sublingual nitroglycerin were administered prior to the scan if clinically indicated. Prospective and retrospective gating protocols were utilized with a tube voltage varying between 70–150 kV. Reconstructed matrix size was $512 \times 512$ with a pixel size between 0.289–0.496 mm. Slice thickness and increment were 0.6 mm and 0.4 mm, respectively.

The secondary dataset included 53 patients. It was retrospectively collected from patients who underwent contrast enhanced CCTA from an independent test center. CCTA was acquired on a dual-source CT scanner (SOMATOM Force, Siemens Healthcare GmbH, Erlangen, Germany). Beta-blockers were not used since physiological cardiac function was also assessed. Tube voltages for the scans varied between 80–120 kV. Reconstructed matrix size was $512 \times 512$ with a pixel size of 0.391 mm. Slice thickness and increment were 0.75 mm and 0.4 mm, respectively.

### 2.2. CCTA Annotations

Three expert readers performed lumen annotations. These experts received 25 h of theoretical and practical training from a level 3 SCCT certified cardiothoracic imaging radiologist. Lumen annotations were performed on standard Windows workstations with a dedicated in-house annotation tool. The tool has two orthogonal curved multiplanar reconstruction (cMPR) views and one cross-sectional view where the experts can perform drawing and editing. First, coronary centerlines and corresponding lumen boundaries were generated using previously developed methods [17,18]. These automatically generated centerlines and lumen boundaries were then manually edited to obtain the final lumen annotations. Lumen annotations were performed only between the proximal and distal lesion markers defined by the expert readers before starting the lumen annotation process. To this end, annotators placed markers at the start and end of the diseased coronary artery regions to define lesions for every branch. These lesions were then extended proximally and distally by 5 mm to include healthy coronary artery regions. If the lesions overlapped after the extension, the lesions were merged. For each extended lesion, the experts first checked and edited the lumen boundaries in 4 cMPR views. They also reviewed their results in the cross-sectional view and edited the contours if required. The window and level were automatically set using values extracted from the DICOM tags; however, the experts were encouraged to modify these values according to existing guidelines [19] to achieve the best visualization of the coronary lumen.

### 2.3. Data Preparation for Convolutional Neural Networks

For each extended lesion, the centerline is sampled equidistantly at 0.25 mm intervals. Unit vectors that are tangential to the centerline are computed. To define a 2D local coordinate system along the centerline, two other unit vectors are determined at every centerline point using a rotation minimizing frame technique [20]. Cross-sectional images are then generated by sampling the CCTA volume at regular grid positions around the centerline along the local 2D coordinate system. Distances from the corresponding branch mesh to regular grid positions are also computed to generate corresponding cross-sectional distance maps, which then can be binarized to obtain lumen masks. The resulting cross sectional images and masks have a size of $64 \times 64$ pixels with 0.125 mm isotropic pixel spacing.

### 2.4. NF Architectures

We investigated the use of a Glow-style NF architecture, combining layers previously introduced in [10,11], such as checkerboard and channel masking coupling layers, invertible $1 \times 1$ Convolutions, Split and Squeeze layers. Our baseline network is depicted in Figure 1 and described in Table 1. We employed affine coupling layers as in Equation (1), where $x$ and $y$ are the input and output tensors, respectively. Subscripts $a$ and $b$ typically denote the two halves of the tensors: one which is passed-through unchanged and the other one which is updated in a linear fashion with respect to itself, but in a highly non-linear fashion with respect to the former half, through functions $s$ and $t$ (which are Deep Neural Networks).

$$
\begin{aligned}
y_a &= x_a \\
y_b &= (x_b - t_{DNN}(x_a)) \, s_{DNN}(x_a)
\end{aligned}
\tag{1}
$$

Networks $s$ and $t$ are in our case a two-head 3D CNN, with its architecture described in Table 2. The final activation function of head $s$ was chosen as $\exp(\tanh(x))$ in order to easily compute the contribution to logDet (as $\sum \tanh(x)$ across all spatial dimensions and channels) and provide a bound of $[e^{-1}, e^1]$ to the scaling done at each coupling layer, ensuring numerical stability and a bounded global maximal value of logDet.

The input samples consist of chunks of 8 adjacent cross sections (down-sampled to $32 \times 32$ resolution) and 2 channels (the concatenation of the angiography and the binary mask volumes). There are 3 squeezing operations which contract the input resolution

$2^3$ times down to $1 \times 4 \times 4$, with increasing number of channels, being the same setup as used in Glow [11] (however we employed fewer layers on each scale due to runtime considerations). The effective receptive field of a coupling layer is given by the receptive field of the $s$ and $t$ network, in this case, $5 \times 5 \times 5$. Stacking coupling layers and using multiple scales (i.e., squeeze layers) increase the final NF receptive field, similar to the operation of classical CNNs.

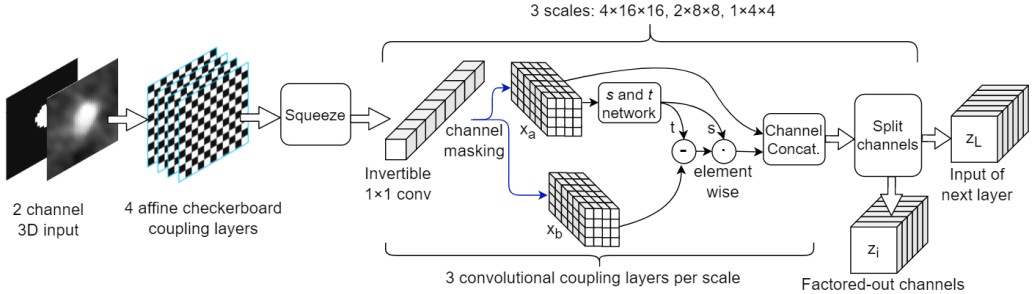

**Figure 1.** Baseline model architecture. Activation norm not depicted.

**Table 1.** Glow-style baseline architecture.

| Stage | No. Blocks | Block Description | Resolution | No. Channels | Total Number of Parameters |
|---|---|---|---|---|---|
| 1 | 4 | Affine coupling layer using checkerboard mask; Activation Norm (if not last block) | $8 \times 32 \times 32$ | 2 | ~2 millions |
| 2<br>3<br>4 | 1 | 3D Squeeze operation | Stage 2: $4 \times 16 \times 16$<br>Stage 3: $2 \times 8 \times 8$<br>Stage 4: $1 \times 4 \times 4$ | Stage 2: 16<br>Stage 3: 64<br>Stage 4: 256 | |
| | 3 | Activation Norm; Invertible $1 \times 1$ Convolution; Affine coupling layer using channel-wise masking | | | |
| | 1 | Split channels | | After stage 2: 8<br>After stage 3: 32 | |

**Table 2.** $s$ and $t$ network architecture.

| Stage | Block | No. Filters | Cumulative Receptive Field |
|---|---|---|---|
| 1 | Conv3D with $3 \times 3 \times 3$ kernel, stride 1, padding 1; BatchNorm; LeakyReLU | 64 | $3 \times 3 \times 3$ |
| 2 | Conv3D with $1 \times 1 \times 1$ kernel, stride 1, padding 0; BatchNorm; LeakyReLU; Dropout | 64 | $3 \times 3 \times 3$ |
| 3 | Conv3D with $3 \times 3 \times 3$ kernel, stride 1, padding 1; BatchNorm; LeakyReLU; Dropout | 64 | $5 \times 5 \times 5$ |
| $4 - s$ | Conv3D with $1 \times 1 \times 1$ kernel, stride 1, padding 0 | As many as $x$'s channels for checkerboard masking or half for channel masking | $5 \times 5 \times 5$ |
| $4 - t$ | Conv3D with $1 \times 1 \times 1$ kernel, stride 1, padding 0 | | |

In [13], it has been shown that NFs which employ affine coupling layers are prone to focus more on local pixel correlations instead of semantic content and exploit coupling layer co-adaptation in order to maximize the final log-probability, i.e., the inductive bias of a stack of affine coupling layers encourages them to encode information about masked pixels in subsequent layers so that $t$ is a good approximation to $x_b$ and thus $s$ can be increased,

leading to larger final log-probabilities. Therefore, stacks of affine coupling layers are incentivized to guess local pixel values by exploiting texture correlations and information feed-forward by bypassing the masks, instead of building features increasing in complexity and expressivity as it is happening for classical stacks of convolutional layers. Semantic features can describe higher level characteristics of the modeled objects, such as global shapes (e.g., the relatively circular shape of the lumen with its spatial continuity between slices), global appearance (e.g., how distinguishable is the lumen from the background) and object correlations (e.g., the mask's spatial alignment to the lumen; calcifications, if present, should be around the lumen, etc.).

Behrmann et al. [21] shows that by constraining a general-purpose residual network [22] to be bi-Lipschitz, it can be used as a NF architecture. The expressive power of ResNets is shown to be preserved even with this constraint. On a classification task the invertible ResNet performed better while on a density modeling task it performed similar to affine-coupling networks. However, sampling from a NF consisting of a ResNet is an iterative process at each residual layer and training/inference involves approximating the logDet at each layer through a power series truncation. Therefore, such ResNet-based models are not that straight-forward in their operation as, e.g., Glow-based models. Ref. [23,24] tackle the problem of conditional probability modelling through the use of NF. In coupling layers, instead of using only the passed-through portion of the layer input to compute $s$ and $t$, an additional embedding (dependent on the conditioning variable) is also employed. In addition, several layers such as Activation Norm (ActNorm) and invertible $1 \times 1$ Convolutions no longer have constant (but trainable) parameters for all inputs/samples, but instead their weights are computed based on the conditional embedding, therefore tailoring their effect for each particular pair of (condition, sample).

Inspired by the above research, we propose the use of a novel type of coupling layer, one which can operate efficiently for both NF directions, does not focus on local pixel correlations and has an inductive bias similar to conventional CNNs. The layer resembles a standard Glow-like sequence of $1 \times 1$ Invertible Convolution, channel masking, affine coupling layer. However, the last step is replaced with a $1 \times 1$ convolution (with applied bias) whose parameters are computed based on the passed-through channels, as in [25]. The applied bias is broadcasted to all spatial positions, therefore is it the same across the width, height and depth of the resulting tensor, meaning that the layer is no longer capable to reproduce masked pixel values as revealed in [13]. The same (sample specific) convolution kernel is applied at all spatial positions, in contrast to the element-wise computation done in (1). This behavior is similar to classical CNNs, with the exception that now the filter weights are not the same for all samples. Equation (2) describes the layer's operation, with simplified notation: $*$ means $1 \times 1$ Convolution with kernel $k$ and $+$ is a broadcasting sum. $k$ is computed by a CNN and has shape $c_{modif}$-by-$c_{modif}$, where $c_{modif}$ is the number of channels which are updated. $b$ is a vector of $c_{modif}$ elements. The CNN responsible for computing $k$ and $b$ is described in Table 3.

$$
\begin{aligned}
y_a &= x_a \\
y_b &= x_a * k(x_a) + b(x_a)
\end{aligned}
\tag{2}
$$

It is observed that the layer is self-conditioned, i.e., it does not employ an external conditioning network or another parallel flow as in [23,24], since the lumen binary mask and the angiographic image were not treated separately, but were concatenated on the channel axis. This is possible because the mask and the image should be highly correlated spatially in order to achieve high log-probability.

A new NF architecture was designed employing the above coupling layer. The first stage is a sequence of Additive Coupling Layers with checkerboard masking. According to [13], these layers will focus mainly on local pixel correlations, but this is equivalent to the functioning of the first layers in classical CNNs, where the receptive field-of-view is small and the filters tend to search for simple patterns such as corners, edges, textures, etc. As

opposed to affine couplings, additive couplings are volume preserving, i.e., they do not contribute directly to logDet and final $\log(p(x))$, but indirectly through the upstream layers.

**Table 3.** CNN architecture for computation of $k$ and $b$ employed inside the coupling layer.

| Stage | Block | No. Filters | Cumulative Receptive Field |
|---|---|---|---|
| 1 | Conv3D with $3 \times 3 \times 3$ kernel, stride 1, padding 1; BatchNorm; LeakyReLU | 64 | $3 \times 3 \times 3$ |
| 2 | MaxPool3D $2 \times 2 \times 2$, stride 2 | | $4 \times 4 \times 4$ |
| 3 | Conv3D with $1 \times 1 \times 1$ kernel, stride 1, padding 0; BatchNorm; LeakyReLU; Dropout | 64 | $4 \times 4 \times 4$ |
| 4 | Conv3D with $3 \times 3 \times 3$ kernel, stride 1, padding 1; BatchNorm; LeakyReLU; Dropout | 64 | $8 \times 8 \times 8$ |
| $5 - k$ | Conv3D with $1 \times 1 \times 1$ kernel, stride 1, padding 0; Average pooling | $c_{modif}^2$ | full |
| $5 - b$ | Conv3D with $1 \times 1 \times 1$ kernel, stride 1, padding 0; Average pooling | $c_{modif}$ | full |

The next stages consist of cascades of coupling layers, as described in Figure 2 and Table 4. In contrast to a classical CNN, where filters of shape $3 \times 3$ (or larger) and strides larger than 1 are used (either in convolutional or max pool layers) to increase the effective field-of-view (FoV), in our architecture the FoV in these stages is increased solely by the squeeze operations. After squeezing, a $1 \times 1 \times 1$ patch of pixels is formed from a patch of $2 \times 2 \times 2$ pixels which were flattened spatially into the channel dimension. Therefore, the FoV doubles on each spatial axis for each squeeze step. This allows $1 \times 1$ Convolutions to operate on increasingly larger FoV, similar to the functioning of a classical CNN, while still retaining the capability of efficient forward/backward NF computation. There are enough squeeze operations so that the resolution on the last stage decays to $1 \times 1 \times 1$. Naturally, we restrict the input spatial dimensions to be powers of 2.

One possible disadvantage is that after each squeeze operation, the number of channels $c_i$ (at stage $i$) increases exponentially with the number of squeezed dimensions (see Table 4). This directly impacts the proposed coupling layer's runtime and complexity, since it must produce matrix $k$ whose size scales with the square of $c_i$. In addition, inference and sampling involve computing the determinant and inverse of $k$, respectively. One workaround to alleviate this issue is to modify the splitting layers so that the tensors are not split in half along the channel axis anymore, but instead only a quarter is retained for the rest of the computation graph while the other 75% of channels are factored out. This can be applied especially in the first stages, where the embeddings mostly describe texture. After such a split, the input to the next squeeze has only $c_i/4$ channels, half that of a regular split. Cascading such splits throughout the network can alleviate the effect of the exponentially-increasing $c_i$, especially for larger resolution inputs. In our experiments, the first split layer only retains $c_i/4$ channels. The net effect is that there are only 512 channels in the last stage, as opposed to the original 1024 (as described in Table 4), resulting in faster runtimes and fewer model parameters.

In this new architecture, BatchNorm was employed instead of ActNorm. In classical CNNs, batch norm acts by computing the batch statistics and then using them to normalize the output. In our approach, two running averages of the batch mean and standard-deviation are employed for normalization and they are updated with current batch statistics after their use, so that the normalization procedure is dependent only on past batches and any cross-talk between samples in the current batch is eliminated. In either CNN and NF cases, batch norm's main purpose is to provide "checkpoints" for activations inside the network, i.e., after each BatchNorm layer the activations have preset statistics (i.e., are

centered around 0 with a std. dev. of 1). This has been shown to improve the training process [11,26].

In all our experiments, the network weights are initialized such that the layers are an identity mapping in the beginning of training, as suggested in [11]. We employed the PyTorch DL framework with the Adam optimizer with a learning rate of $1 \times 10^{-4}$ and trained until the validation loss plateaued.

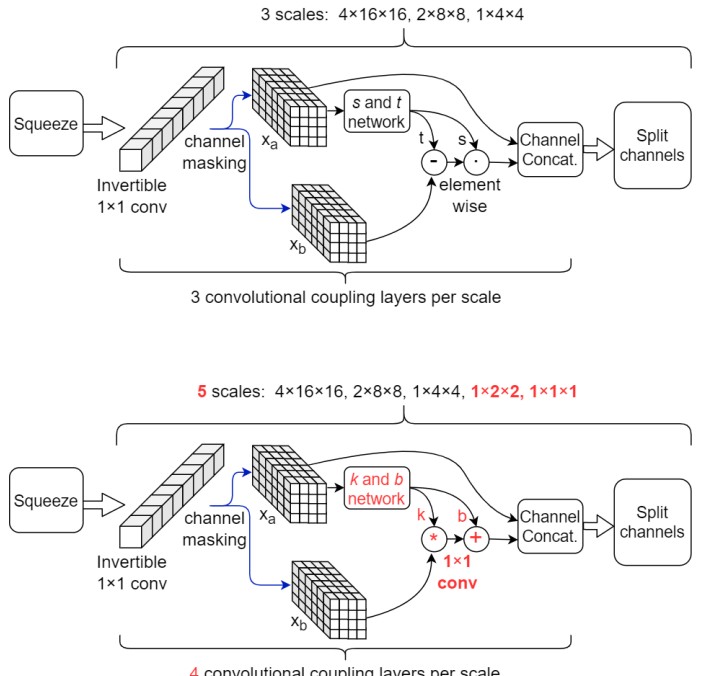

**Figure 2.** Comparison between baseline inner architecture (**top**) and proposed inner architecture (**bottom**) employing the novel coupling layer. Updated parts are highlighted in red. Normalization layers not depicted.

**Table 4.** Improved NF architecture employing the novel coupling layers.

| Stage | No. Blocks | Block Description | Resolution | No. Channels | Total Number of Parameters |
|---|---|---|---|---|---|
| 1 | 4 | Additive coupling layer using checkerboard mask; BatchNorm (if not last block) | $8 \times 32 \times 32$ | 2 | |
| 2 3 4 5 6 | 1 | 3D Squeeze operation | Stage 2: $4 \times 16 \times 16$ Stage 3: $2 \times 8 \times 8$ Stage 4: $1 \times 4 \times 4$ Stage 5: $1 \times 2 \times 2$ Stage 6: $1 \times 1 \times 1$ | Stage 2: 16 Stage 3: 64 Stage 4: 256 Stage 5: 512 Stage 6: 1024 | ~8.7 millions |
| | 4 | BatchNorm; Invertible $1 \times 1$ Convolution; convolutional coupling layer using channel-wise masking | | | |
| | 1 | Split channels | | After stage 2: 8 After stage 3: 32 After stage 4: 128 After stage 5: 256 | |

## 2.5. Synthetic Mask Perturbations

Our application's goal is to detect incorrect pairs of (angiography image, lumen mask), i.e., samples where the segmentation is not in full agreement with the image. To test our models, we devised a method to obtain "wrong" datapoints (or samples which are not in the distribution of "correct" image-mask pairs) starting from our initial data (considered to be "correct").

We augmented the datasets by applying preset perturbations on the lumen segmentation binary mask, while keeping the angiographic image untouched. Three types of mask perturbations were employed:

- zooming:we applied zoom in/out operations on the mask image with respect to the mask center, so that the resulting mask is still aligned with the angiography, but larger/smaller than before. Figure 3 displays an example for various levels of zoom.
- morphing: we applied dilations or erosions along 4 directions on the height * width plane: left-right, top-bottom, topLeft-bottomRight and topRight-bottomLeft. This perturbation only affects one part of the mask (the eroded or dilated part), while the other part is left untouched. Figure 4 displays an example for various levels of morphing. By convention, negative and positive levels refer to the two ways in the selected direction, with zero meaning original mask position (levels are expressed as ratios of the original mask size along the chosen direction). At every level, either dilation (resulting in prolonged masks) or erosion (resulting in shortened masks) can be applied.
- translations: in the same 4 directions on the height * width plane, we translated whole mask images. Each level increment signifies a pixel shift. Figure 5 shows an example for various levels of translation.

For each network architecture, we performed two training procedures: one employing only original (unperturbed) data and one employing a dataset consisting of the original data and its perturbed version. The perturbations are applied during train time, similar to data augmentation techniques, such that each original data sample gets perturbed on all perturbation types, levels and directions over the training epochs. At each epoch, the ratio between untouched and perturbed data is 1-to-1.

When only original data is used, the training loss function consists of maximizing the log-probabilities across the train set. When perturbed data is also employed, we used a train loss function (Equation (3)) similar to the hinge loss introduced in [13,27], where the model tries to maximize predicted log-probs for original (untouched, *in-distrib*) samples and tries to decrease predicted log-probs under a certain threshold $T$ for perturbed samples (OoD). The hinge loss allows us to shape the learnt probability density modeled by the NF, by directly offering supervision in regions around the original samples in the input domain. Therefore, the NF can be trained to be sensitive to the used synthetic perturbations and to mark perturbed samples as OoD

$$L(\theta, x) = \mathbb{E}_{x \in inDistrib}(\ln(p_\theta(x)) - \mathbb{E}_{x \in \text{OoD}}(\max(0, \ln(p_\theta(x)) - T))$$
$$\theta_{optimal} = \text{argmax}_\theta \, L(\theta, x)$$

(3)

where $p_\theta$ is the probability density modeled by the NF.

Training only on original data and then testing on synthetic perturbations gives insight into the OoD detection capability which stems purely from the inductive bias of the NF architecture. In addition, we argue that NF models, being a class of generative models, provide a form of explainability by being able to produce samples from their learnt probability density. By sampling repeatedly from the model and computing the associated log-probs, one can observe the kind of samples which the model considers to be in-distribution. We believe that this gives insight into the semantic content which is interpreted by the model and into the functioning of the computational chain of layers. Section 3.3 will discuss in further detail.

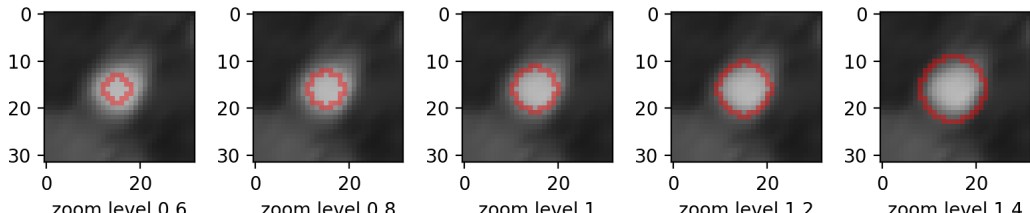

**Figure 3.** Example of zoom perturbation on lumen mask. Only the mask contour is shown.

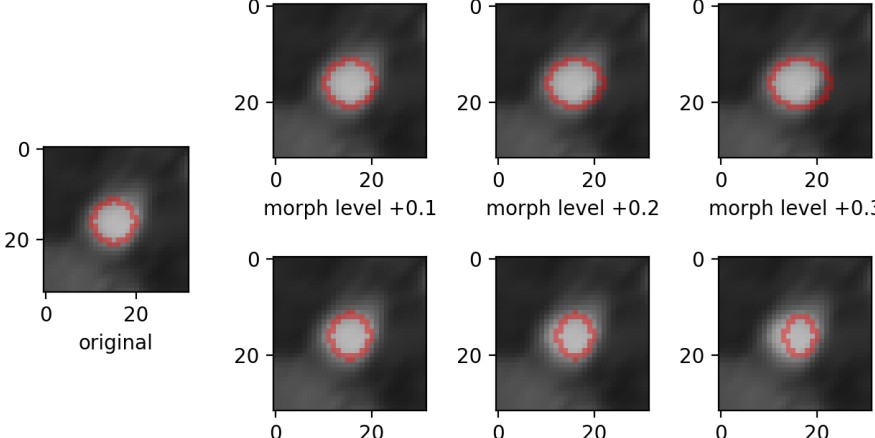

**Figure 4.** Example of mask morphing perturbation in the left-to-right direction. Top row shows dilations (the mask is prolonged), bottom row shows erosions (the mask is shortened). Only the mask contour is shown.



**Figure 5.** Example of mask translation perturbation in the topLeft-bottomRight direction. Only the mask contour is shown.

## 3. Results and Discussion

### 3.1. Evaluation on Synthetic Mask Perturbations

First, we evaluated the baseline and the proposed networks trained on original (unperturbed) data. We applied the synthetic perturbations on the testset in increasing levels of severity and measured how well the models can distinguish between log-probs of original and log-probs of perturbed samples. At each perturbation level, we computed the area under the RoC curve. Figures 6–8 display the AUROC values for translation, zooming and morphing perturbations, respectively. We use AuRoC as a metric for assessing how well two individual data distributions can be separated by using a probability threshold. A value close to 1 indicates that there are probability thresholds which yield near 100% accuracy in detecting perturbations, while values close to 0.5 indicate that the probabilities of the two data distributions have high overlap and are therefore indistinguishable by simple thresholding. Hence, the closer the AuRoC values are to 1, the better is the performance of the method. Zoom level $1.0\times$ and translate level 0 do not have any effect on the test data, so naturally the AuRoC is 0.5 since it is comparing the same distribution against itself.

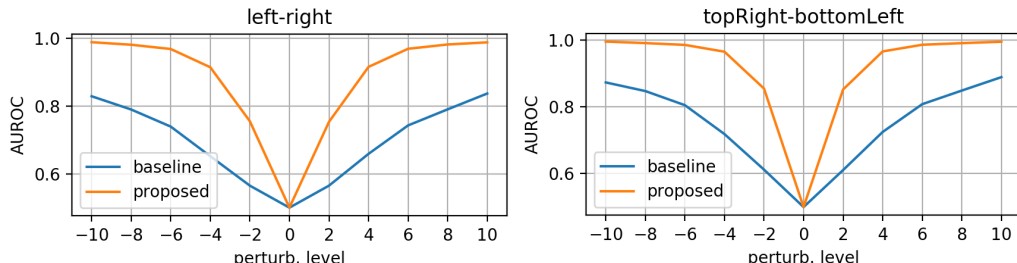

**Figure 6.** AuRoC performance for the baseline and proposed architecture when tested against mask translation in various directions. Training done only on original data.

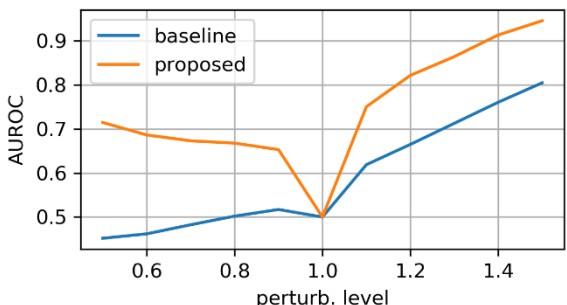

**Figure 7.** AuRoC performance for the baseline and proposed architecture when tested against mask zooming. Training done only on original data.

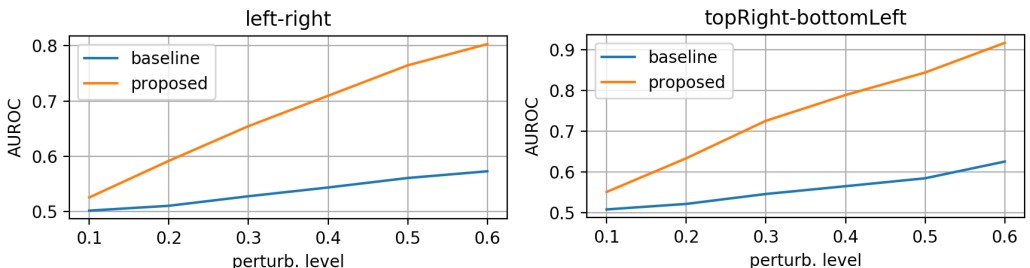

**Figure 8.** AuRoC performance for the baseline and proposed architecture when tested against mask morphing in various directions. Training done only on original data.

It is observed that the proposed model has superior performance across all perturbation types and levels. Zooming under $1.0\times$ actually yields higher log-probs for the baseline model, resulting in AuRoC values under 0.5. Even at small mask perturbation levels (e.g., $0.9\times/1.1\times$ zooming, $\pm 2$ pixels translation), the proposed model has much larger sensitivity in detecting the mask alterations (even though it was not trained explicitly to do so) in contrast to the baseline model, where the log-probs start to decrease more significantly only at larger perturbation levels. The mask morphing is the hardest to detect since part of the mask remains the same. Thus, the baseline model is largely insensitive to this type of perturbation as the maximum AuRoC at a high perturbation level of 60% is under 0.65. In comparison, the AuRoC for the proposed network has a much faster variation for increasing perturbation severity, achieving values over 0.9 for some directions at 60% morphing.

Next, we evaluated the test-time sensitivity against synthetic perturbations after training using the augmented trainset and loss from Equation (3). We employed the following perturbation levels to generate OoD samples for training:

- translation (in all 4 directions) of $\pm 3$ or $\pm 4$ pixels;
- zooming levels of $0.65\times$, $0.8\times$, $1.2\times$, and $1.35\times$;
- morphing (in all 4 directions, erosions/dilations) levels of 0.2 and 0.35 (ratio of initial mask size).

Any OoD sample had only one type of perturbation applied to it. In addition, given the fact that each sample is a 3D volume consisting of 8 2D slices, the perturbation could be applied on each slice following some preset variation for its severity/level along the slice axis (see Figure 9 for the employed severity variation types). A severity variation type is sampled randomly for each OoD sample and each component 2D slice is perturbed according to its corresponding severity.

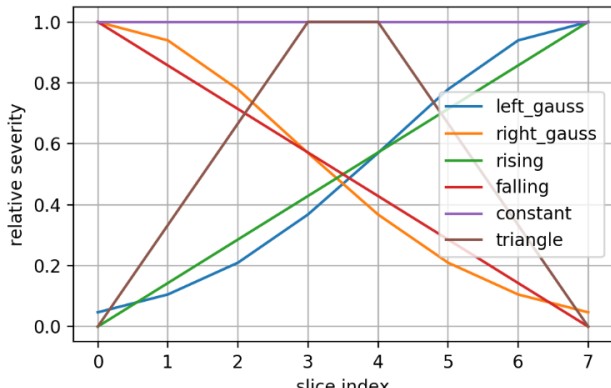

**Figure 9.** There were 7 perturbation severity variations employed for OoD samples generation. Each slice index has its corresponding relative severity level, according to the chosen severity variation.

Figures 10–12 display the test-set AUROC values for translation, zooming and morphing perturbations, respectively. The two models perform similarly well, except for some morphing directions, where the proposed model has slightly lower AuRoC values for small perturbation levels.

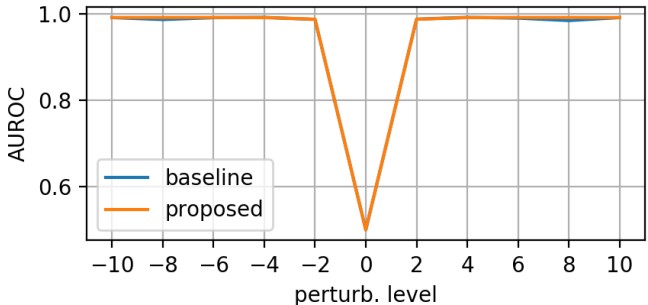

**Figure 10.** AuRoC performance for the baseline and proposed architecture when tested against mask translation in various directions. Training done on augmented dataset using the hinge loss in Equation (3).

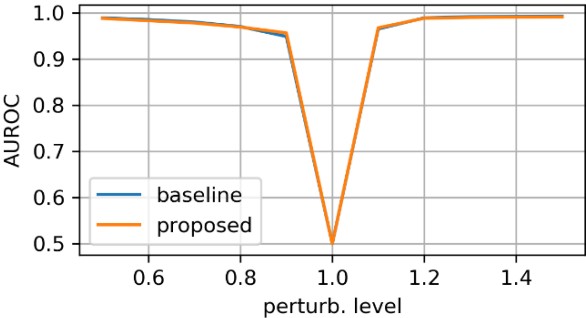

**Figure 11.** AuRoC performance for the baseline and proposed architecture when tested against mask zooming. Training done on augmented dataset using the hinge loss in Equation (3).

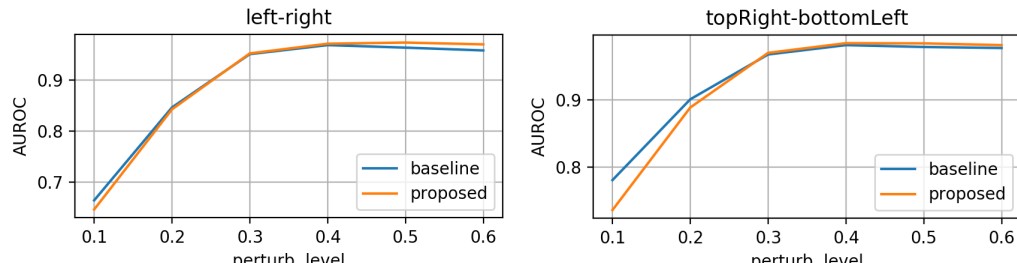

**Figure 12.** AuRoC performance for the baseline and proposed architecture when tested against mask morphing in various directions. Training done on augmented dataset using the hinge loss in Equation (3).

This training procedure yields models which show high sensitivity towards samples from the outlier distribution used explicitly during training, and which can separate the log-prob distributions even for small levels of perturbations. However, in [13] it is reported that even though NF models can achieve good separation between in-distrib and OoD sets used explicitly during training, there may also be other OoD sets (unseen during training) which still achieve log-probs as high as in-distribution data. Therefore, the inductive bias of the model still plays a major role in the generalization and usability of a NF model in the face of new data, even if it was explicitly tuned to decrease log-probabilities for *some* forms of outliers. The next sections will further inspect the two models trained on augmented data and will provide evidence that the baseline model, even though it can now detect some outliers, does not model a useful probability density which is descriptive of the training data.

To obtain a log-probability signal which describes the likelihood of an entire vessel segment, a sliding window approach was employed in which overlapping chunks of 8 adjacent cross-sections are fed through the NF model to obtain the log-prob values for each chunk. Using this procedure, middle cross-sections can participate in at most 8 chunks, therefore there may be up to eight predicted log-probability values linked to each middle cross-Section. A voting scheme based on averaging is employed, where the final log-prob value for each cross-section is computed by averaging the linked predicted log-probs. Figure 13 depicts such an example, where a synthetic perturbation is applied with a known severity variation. The proposed NF model detects when the perturbation severity is high enough, while outputting high log-prob values when the perturbation is negligible.

### 3.2. Evaluation on Expert Annotations

The results in the previous section indicate that the herein proposed model is superior to the Glow-like baseline. Hence, we first ran the proposed model (trained on the augmented trainset) on the secondary dataset described in Section 2.1. We employed two relative thresholds of 60% and 90% of the mean log-probability value observed on the primary dataset's test split, when no perturbation was applied. All lesions which had *at least one* cross-Section log-probability value under the 60% threshold were selected as candidates with possibly wrong mask annotations, yielding a total of 31 of these lesions. To construct the bin of candidates with possibly correct mask annotations, we randomly sampled 31 lesions from those for which all cross-Section log-probability values were above the 90% threshold. Thus, we did not consider lesions which had any intermediate values of log-probability (i.e., between the 60% and 90% thresholds) without also having at least one low log-probability cross-section, to avoid the effect of model uncertainty for data which it considers to be near the separation manifold between correctly annotated lesions and faulty ones. We then further excluded lesions which had a reference diameter (computed as average of healthy proximal and distal diameters) lower than 1.5 mm (the typical threshold employed in CCTA based studies assessing CAD anatomically and functionally). As a result, a test set containing 56 lesions from 35 patients was employed for the evaluation.

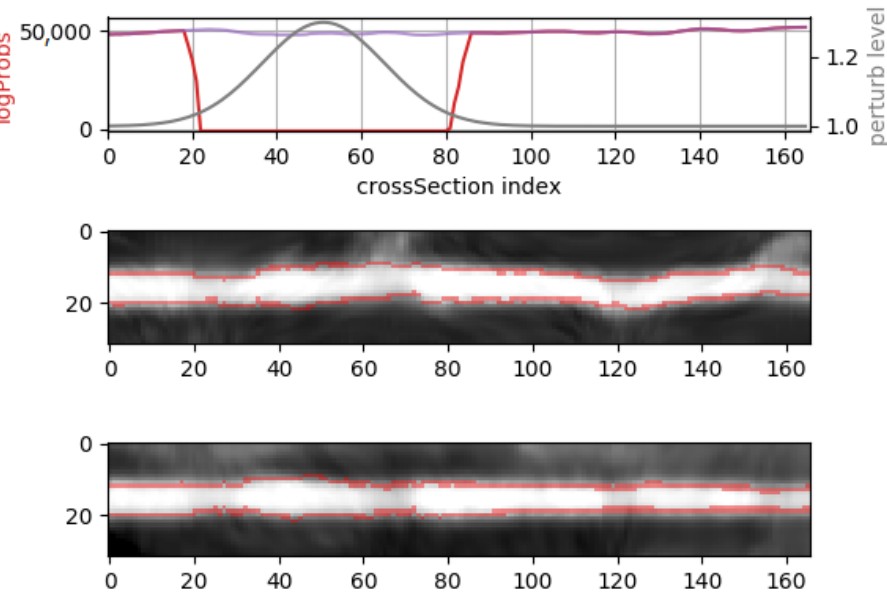

**Figure 13.** Whole vessel-segment prediction using a sliding window approach and the proposed architecture. A zooming perturbation with a known severity variation (top plot, gray signal) is applied (note that zoom level 1.0× is an identity transform). The resulting log-prob signal (top plot, red signal) dips whenever the perturbation is severe enough, compared to the original log-prob signal in the absence of any perturbation (top plot, purple signal). The bottom 2 plots display two lateral views of the vessel segment (2 projections on different axes), with the perturbed mask contour overlaid.

The selected test set was manually and independently annotated by three expert readers at lesion-level: each lesion was marked as being either "correctly" or "incorrectly annotated", based on the following instructions: a lesion should be marked as "incorrectly annotated" if *at least one* cross-sectional contour would require editing; otherwise, if *no* cross-sectional contour requires editing, then the lesion should be marked as "correctly annotated". The annotator instructions were devised so as to match the procedure employed to construct this separate test set, with the goal of being able to directly compare the labels from the NF model to the ones provided by the annotators.

The Glow-like baseline NF model was also applied on the 56 lesions and the same relative thresholds and criteria were employed to classify each lesion. Evaluating the two NF models against the human annotations was framed as a binary classification problem. Table 5 summarizes relevant metrics (accuracy, sensitivity, specificity, PPV and NPV) for the proposed and baseline models. Annotation consensus was obtained through a majority vote between the three annotators. The mean inter-user metric values were obtained by averaging all 6 possible metric values pertaining to pairs of annotators, e.g., Annotator_1 (as GT) versus Annotator_3 (as Prediction), Annotator_3 (as GT) versus Annotator_1 (as Prediction), etc. When compared against annotation consensus, the proposed model has higher performance than the baseline on all considered metrics.

Of special interest is the sensitivity metric, which measures the percentage of NF-flagged lesions as being incorrect from the set of lesions considered incorrect by the human annotators' consensus. The higher this metric value, the more capable is a NF model in detecting faulty segmentation masks. We observe that the proposed model has sensitivity of 76.0%, close to the inter-user value of 79.0%, while the baseline model only achieves 48%. Overall, according to the majority vote of the expert readers, 25 lesions were annotated as requiring editing, out of which 17 had unanimous annotations and 8 had non-unanimous annotations. The NF model correctly classified 16 out of the 17 unanimously annotated

lesions and three of the eight non-unanimously annotated lesions. This indicates a 94.1% sensitivity on the unanimously annotated lesions.

The overall accuracy score also increases to 78.6% (close to the inter-user value of 80.9%) for the proposed model, as compared to an accuracy value of 64.3% for the baseline Glow-like model. These results reinforce the observation that the baseline model is unable to fully capture semantic content while the proposed model does. Similar behavior was observed in the previous section, where the proposed model had better AuRoC values in detecting synthetic perturbations when trained only on original data.

**Table 5.** Metrics on the secondary dataset for the baseline and the proposed model. The proposed model consistently outperforms the baseline and has metric values close to inter-expert agreement.

| Metric | Inter-Expert Agreement Average [Min, Max] | Baseline Model | Proposed Model |
|---|---|---|---|
| Accuracy | 0.81 [0.79, 0.86] | 0.64 | 0.79 |
| Sensitivity | 0.79 [0.70, 0.87] | 0.48 | 0.76 |
| Specificity | 0.83 [0.76, 0.90] | 0.77 | 0.81 |
| PPV | 0.79 [0.70, 0.87] | 0.63 | 0.76 |
| NPV | 0.83 [0.76, 0.90] | 0.65 | 0.81 |

*3.3. Sampling from the Models*

We employed the models trained on the augmented trainset to generate novel samples. Similar to sampling procedures in [11], we employed $\mathcal{N}(\mathbf{0},\ 0.6 \cdot I)$ instead of the actual prior distribution (i.e., standard normal multivariate distribution) in order to produce samples with larger log-probs and which look more realistic. Each new sample was run back through the model in the forward direction to compute the log-probs, confirming that the sample is in fact in-distribution (the sampling procedure may seldomly generate samples of lower log-probability). Figure 14 shows samples from the two models.

As already revealed in [13], the baseline model tends to focus more on textures and is unable to capture the semantics of the training data. We observed that in most of the generated samples, the segmentation mask is lacking (i.e., only zeros are generated on the mask channel). In addition, the usual round shape of the lumen is not distinguishable in the image channel. In [11], the proposed Glow model can indeed generate realistic samples. That model operates on the same spatial resolution as ours ($32 \times 32$) and uses the same number of 3 spatial scales (i.e., squeeze operations); however, it employs up to 48 coupling layers per scale (as opposed to ours, which only uses 3 coupling layers per scale due to runtime considerations). We hypothesize that many glow-like layers are required at each scale because of their tendency to disregard semantic content and a deep stack of such layers can approximate some semantic content as very complex textures.

In contrast, the proposed architecture uses a small number of 4 (novel) coupling layers per scale and manages to capture the semantic content of a usual data point: the lumen has the typical shape in the image channel, the segmentation mask is present (with plausible pixel-values, e.g., close to either 0 or 1) and respects the shape and position of the lumen in the image channel.

We argue that inspecting the generated samples is an explainability mechanism which offers insight into the learnt probability distribution, i.e., the model can provide example inputs which are very likely under the learnt density and by repeating the sampling procedure enough times, an approximation of the typical set of the learnt distribution may be constructed. If a generative model consistently produces samples with high associated log-probabilities but which have low quality under manual inspection and are implausible considering the specific topic/domain, then this is proof that the learnt probability density is not a good approximation of the true probability density and, therefore, the model cannot be reliably used for OoD detection (even if its train-time loss function encouraged the

separation of *some* OoD data sets). In practice, the amount of available OoD data is usually limited and the inductive bias of the model still holds a huge importance in the quality of the distribution fitting. Zhang et al. [27] shows that "even good generators can still exhibit OoD detection failures", therefore the above condition of good/plausible sample generation is necessary, but not sufficient. It is necessary because the learnt probability density should match the underlying data probability density as closely as possible. However, even for a good generator (with high validation-set likelihoods) there may be small-volume regions of the sample space where the model assigns high-density but low overall probability mass. This faulty assignment of high-density might be caused by model estimation error [27], but because the overall probability mass may be negligible (due to the small volumes of poorly modeled regions), it does not affect the generation of synthetic samples. Still, training a generative model for OoD detection can require accurate estimation in regions which are unimportant for good generation [27]. Using an ensemble of generative models may alleviate the effect of model estimation errors, as each model instance may mis-predict on different regions of the sample space and thus errors could be averaged out.

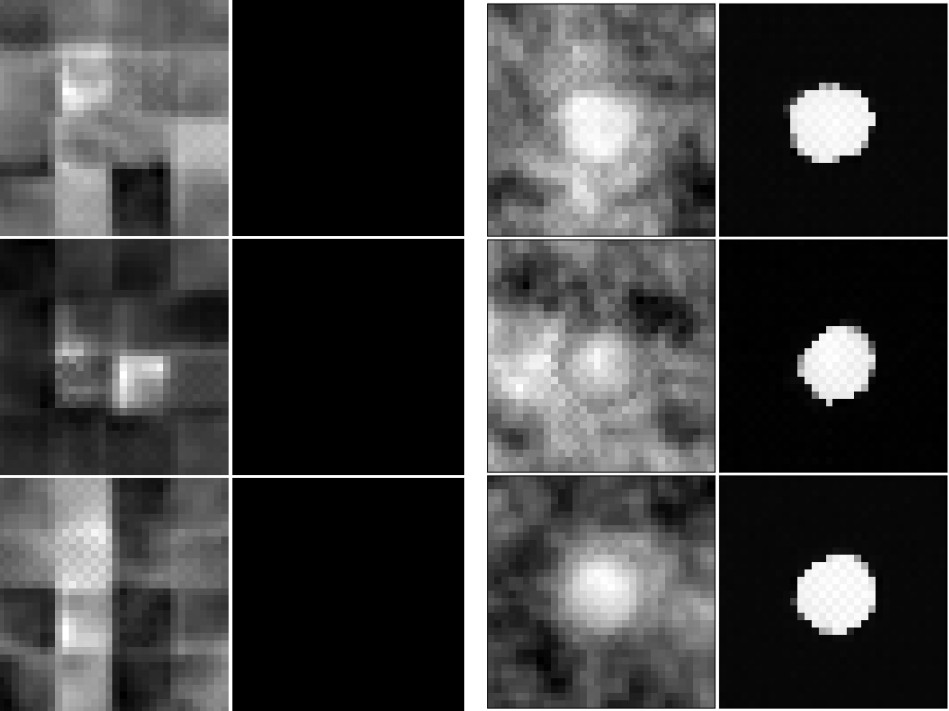

**Figure 14.** Three samples (pairs of lumen image and segmentation mask) generated by the proposed network (**right**) and by the baseline model (**left**).

### 3.4. Inspecting the Flows

The main hypothesis in [28] is that, in hierarchical VAEs, the lowest latent variables "learn generic features that can be used to describe a wide range of data" and thus OoD data can achieve high likelihoods "as long as the learned low-level features are appropriate". It is further suggested that "OOD data are in-distribution with respect to these low-level features, but not with respect to semantic ones".

Inspired by the hierarchical likelihood bounds approach in [28], we inspected the progressive transformation of mask-perturbed samples as they are passed through the sequence of coupling layers inside the NF models (i.e., going in the forward direction from $x \in X$ to $z \in Z$). We recorded the likelihood of the first factored out embeddings (termed $z_{bottom}$) after the first splitting operation and their associated logDet values at that stage. We computed pseudo-likelihood values, associated to these "bottom features", by ignoring the rest of the computational chain and the part of the embeddings which were not factored

out (termed $z_{top}$). We also computed pseudo-likelihood values associated with the "top features", i.e., by summing the likelihood (under the prior) of embeddings $z_{top}$ and the updates to logDet done at stages after the first splitting operation.

Intuitively, the "bottom features" operate at a smaller field-of-view and tend to capture more local patterns, while the "top features" are built based on the "bottom features" and at larger fields-of-views, therefore being able to access the global semantic content of a sample. Formally, the two pseudo-likelihoods are computed as in Equation (4):

$$\begin{aligned}\mathcal{L}_{bottom} &= \ln(p(z_{bottom})) + \ln(|\det(\nabla(f_1 \circ f_2 \circ \ldots \circ f_k))|) \\ \mathcal{L}_{top} &= \ln\big(p\big(z_{top}\big)\big) + \ln(|\det(\nabla(f_{k+1} \circ f_{k+2} \circ \ldots \circ f_M))|)\end{aligned} \tag{4}$$

where $M$ is the number of layers in the NF model and $k$ is the index of the splitting layer which factored out $z_{bottom}$ and retained $z_{top}$ for the upstream layers. Because the chosen prior distribution is a diagonal multivariate Gaussian, the sum of $\mathcal{L}_{bottom}$ and $\mathcal{L}_{top}$ yields exactly $\ln(p(x))$.

Employing models trained only on original (unperturbed) data, we computed AuRoC values for the two pseudo-likelihoods when applying perturbations of increasing severity and compared performances to the standard case where the regular log-probabilities are used to discriminate OoD samples. Figure 15 shows the AuRoC values for the baseline and proposed models when detecting mask translations in various directions. For the baseline model, the bottom features have worse performance in detecting outliers, while the top features perform better than regular log-probs. This observation is in line with the hypothesis in [28], that higher level latent variables can better discriminate through semantic content, while lower level latents would yield similar likelihoods for OoD data if textures appear to be in-distribution.

However, the proposed model shows consistent performance across the top and bottom level features and regular log-probabilities. This suggests that even at the first spatial scale, the novel coupling layers try to capture semantic features instead of local spatial correlations and that the inductive bias of this coupling layer is better suited for OoD detection than regular affine coupling layers (as used in [10] or [11]).

Inspecting the proposed model also on mask-morphing or mask-zooming perturbations reveals the same behavior. However, despite the fact that pseudo-likelihoods of bottom-features exhibit similar OoD detection performance as regular log-probs, the network architecture cannot be truncated to use a smaller number of spatial scales. An experiment where only four spatial scales were employed (instead of the original 5) was conducted. The resolution on the final spatial scale was $1 \times 2 \times 2$ instead of $1 \times 1 \times 1$. The experiment revealed that the network in this configuration is unable to ensure spatial coherency across the entirety of the input image (of resolution $8 \times 32 \times 32$) but only on patches of $8 \times 16 \times 16$, since each pixel position in the $1 \times 2 \times 2$ map has a field-of-view of 16 pixels. Even though the $k$-and-$b$ network has a full receptive view, the $k$ kernel is applied on a $1 \times 1$ basis and cannot semantically link adjacent spatial sections in the $x_b$ tensor. Therefore, the proposed network decays the spatial resolution down to $1 \times 1 \times 1$, where the computed $k$ kernel can operate on the entire receptive field of the input. Figure 16 shows samples generated from a model with only four spatial scales instead of five. The samples reveal that the 4 quadrants are not semantically connected.

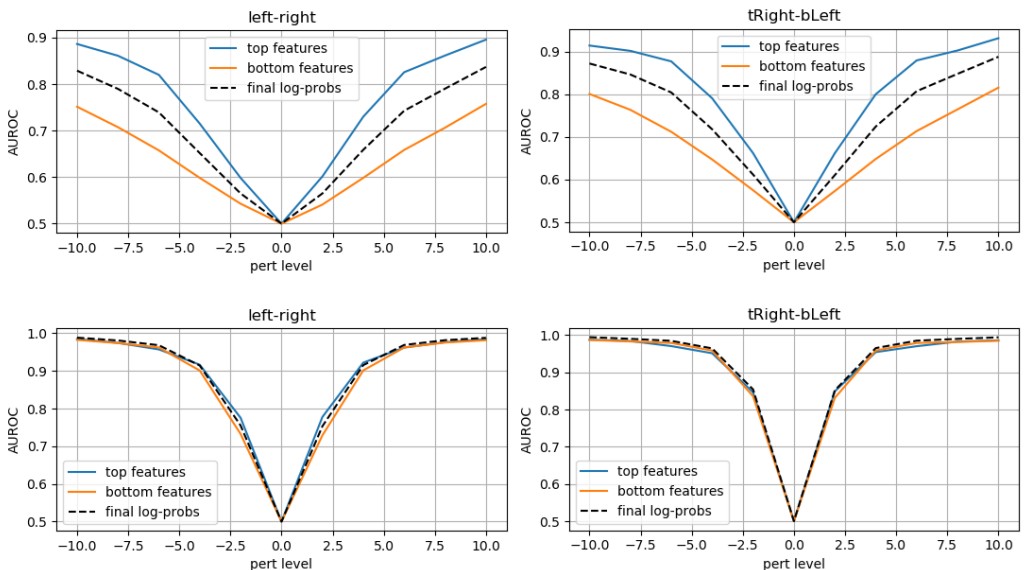

**Figure 15.** Baseline model (**first row**) compared to proposed network (**last row**): AuRoC values for separating OoD mask-translated samples, using either top-features or bottom-features. Dashed black lines represent default model performance using regular log-probs.

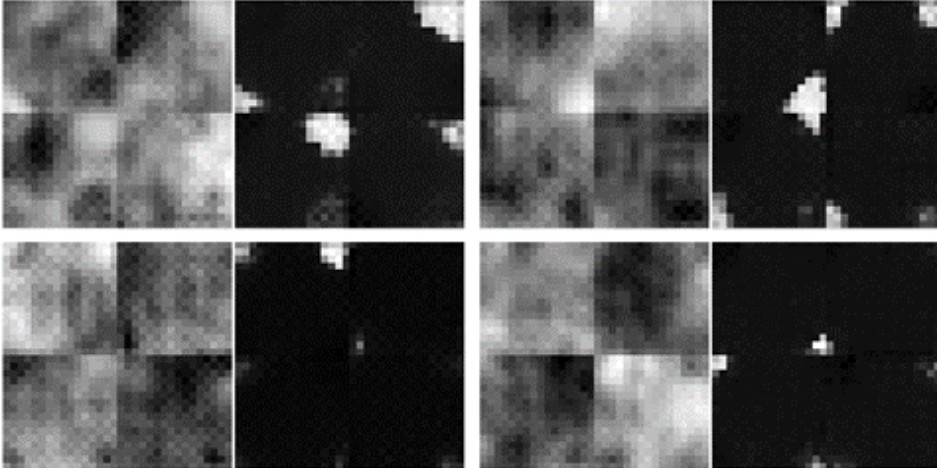

**Figure 16.** Generated samples are not coherent across the entire $32 \times 32$ spatial resolution, but only on the $16 \times 16$ quadrants.

## 4. Conclusions

While the performance of AI based methods has improved markedly over the past few years, semi-automated approaches are currently still being employed. One potential approach for significantly reducing the processing time is to pre-select regions of interest which are likely to require manual inspection and editing. Herein, we linked this preselection step to the topic of confidence and out-of-distribution detection, based on NF. The usage of a novel coupling layer which exhibits an inductive bias favoring the exploitation of semantical features instead of local pixel correlations was investigated on the task of detecting mismatched pairs of CCTA lumen images and their corresponding lumen segmentations. A network architecture employing such layers was tested against a Glow-like baseline. The proposed network showed better performance in OoD detection when tested against synthetic perturbations, while the sensitivity of detecting faulty annotations was close to inter-expert agreement. Samples from the model confirm that the learnt probability density managed to capture the relevant informational content from the training samples, instead of just modelling plain textures.

During model development and testing stages, only manual annotations were employed in the input mask-channel; therefore, the use of the investigated models is not tied to any specific segmentation model. Thus, any model-specific segmentation artefacts, which could possibly alter the observed probability density of correct image-mask pairs, are avoided by learning only from manual annotations (e.g., possible segmentation failure modes are not included in the learnt probability density). In deployment scenarios, the mask channel would be fed by a separate segmentation model and the proposed NF can therefore act as an independent audit model, detecting cases where the mask is not in full agreement with the underlying lumen images. As the failure modes of the NF model would be uncorrelated with the failure modes of the segmentation model, the proposed setup is better suited for robust pre-selection of vessel locations which are likely to require inspection and editing, leading to time savings when performing semi-automated CCTA lumen analysis.

CCTA is a powerful non-invasive test for ruling out CAD, i.e., avoiding unnecessary invasive coronary angiography (ICA). A recent review has summarized the latest aspects addressing the CCTA suitability for selecting patients for invasive coronary angiography (ICA) and subsequent revascularization [29]. Clinical trials have shown that performing CCTA in patients receiving a clinical indication for ICA results in lower costs and more effective patient care [30].

However, small errors in the CCTA interpretation (e.g., minimal lumen area or diameter) can have a significant influence on the interpretation of the anatomical and/or functional significance of a stenosis. A large grey zone of uncertainty in the clinical interpretation may be the consequence.

The method proposed herein allows for more confident decision making using CCTA imaging alone. Using the proposed out-of-domain detection method, the gray zone in the clinical interpretation can potentially be narrowed down.

**Author Contributions:** C.F.C.: Methodology, Software, Validation, Formal analysis, Writing—Original Draft, Visualization. L.M.I.: Conceptualization, Methodology, Writing—Original Draft, Writing—Review & Editing, Visualization, Project administration, Funding acquisition. S.C.: Conceptualization, Investigation, Resources, Data Curation. C.S. and M.W.: Conceptualization, Writing—Review and Editing, Supervision. P.F., S.S., F.A. and S.J.B.: Writing—Review and Editing. P.S.: Conceptualization, Supervision, Project administration, Funding acquisition. S.R.: Conceptualization, Methodology, Supervision, Project administration. All authors have read and agreed to the published version of the manuscript.

**Funding:** The research leading to these results has received funding from the EEA Grants 2014–2021, under Project contract No. 33/2021. This work was partially supported by a grant of the Romanian National Authority for Scientific Research and Innovation, CCCDI–UEFISCDI, project number ERANETPERMED-HeartMed, within PNCDI III.

**Institutional Review Board Statement:** Not applicable.

**Informed Consent Statement:** Not applicable.

**Data Availability Statement:** The data has been acquired as part of the project acknowledged in the manuscript, and cannot be made public, considering GDPR regulations and the content of the informed consent signed by the patients.

**Acknowledgments:** The concepts and information presented in this paper are based on research results that are not commercially available. Future commercial availability cannot be guaranteed.

**Conflicts of Interest:** Costin Florian Ciușdel and Lucian Mihai Itu are employees of Siemens SRL, Advanta, Brasov, Romania. Chris Schwemmer and Michael Wels are employees of Siemens Healthcare GmbH, Computed Tomography-Research & Development, Forchheim, Germany. Serkan Cimen, Puneet Sharma and Saikiran Rapaka are employees of Siemens Healthineers, Digital Technology & Innovation, Princeton NJ, USA. Philipp Fortner, Sebastian Seitz, Florian André and Sebastian Johannes Buß are employees of Das Radiologische Zentrum—Radiology Center, Sinsheim-Eberbach-Walldorf-Heidelberg, Germany.

## Abbreviations

The following abbreviations are used in this manuscript:

| | |
|---|---|
| CCTA | Coronary computed tomography angiography |
| OoD | Out-of-Distribution |
| NF | Normalizing Flows |
| AI | Artificial Intelligence |
| CNN | Convolutional Neural Network |
| GT | Ground truth |
| CAD | Coronary artery disease |
| AuRoC | Area under the Receiver operating Characteristics |
| FFR | Fractional Flow Reserve |
| AHA | American Heart Association |
| CFD | Computational fluid dynamics |
| KL | Kullback–Leibler divergence |
| logDet | Logarithm of determinant |
| cMPR | curved Multiplanar Reconstruction |
| VAE | Variational Auto-Encoder |

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
