# Peer review of "Normalizing Flows for Out-of-Distribution Detection: Application to Coronary Artery Segmentation"

_applsci, doi:10.3390/app12083839_

Round 1
Reviewer 1 Report
Dear authors,
After reading your manuscript I got some unresolved questions, so I would like to ask you about them:
First, what are the semantic features do you point out?
Second, why do you compare your model against base model on generated samples? I mean, knowing it did not produce a segmentation mask.
Third, how much time did you reduce using this approach vs the actual approach.
Thanks in advance for your consideration and time to respond these inquiries.
Best regards,
Reviewer 2 Report
Reviewing the manuscript entitled, “Normalizing Flows for Out-of-Distribution Detection: Application to Coronary Artery Segmentation” by Ciusdel CF et al., this is an article focusing on establishment of CCTA with higher diagnostic accuracy. This is a very important manuscript, and when only less invasive CCTA can make a diagnosis of coronary artery disease, the contribution to medical care will be immeasurable. However, it is a rather esoteric manuscript for the cardiologist who will be the main leader. Therefore, the authors need to respond to the following concerns.
In figure 6 to 12, the authors should describe how to interpret the difference between baseline and proposed architecture of the AuRoC performance for the cardiologist to understand easily.
Accurate diagnosis of coronary artery lesions must lead to optimization of treatment, and if correct images cannot be obtained, coronary angiography is all that is needed. The author should describe the relationship between higher image detection of CCTA that you proposed and coronary angiography as a benefit to the patient who has coronary heart disease in the discussion.
The authors need to add the abbreviation table.
Reviewer 3 Report
Here authors nicely evaluated an approach based on normalizing flows architectures for the Out-of-Distribution detection of coronary lumen segmentations by using a new type of coupling layer.
All controls has been tested, approach is very well described and conclusions are supported by data.
